# COVID-19 Outcomes and Vaccinations in Swedish Solid Organ Transplant Recipients 2020–2021: A Nationwide Multi-Register Comparative Cohort Study

**DOI:** 10.3390/v16020271

**Published:** 2024-02-08

**Authors:** John Mackay Søfteland, Huiqi Li, Jesper M. Magnusson, Susannah Leach, Vanda Friman, Magnus Gisslén, Marie Felldin, Andreas Schult, Kristjan Karason, Seema Baid-Agrawal, Carin Wallquist, Fredrik Nyberg

**Affiliations:** 1The Transplant Institute, Sahlgrenska University Hospital, 41345 Gothenburg, Swedenandreas.schult@vgregion.se (A.S.); seema.baid-agrawal@vgregion.se (S.B.-A.); 2Department of Surgery, Institute of Clinical Sciences, Sahlgrenska Academy, University of Gothenburg, 40530 Gothenburg, Sweden; 3School of Public Health and Community Medicine, Institute of Medicine, Sahlgrenska Academy, University of Gothenburg, 40530 Gothenburg, Sweden; 4Department of Pulmonology, Institute of Medicine, Sahlgrenska Academy, University of Gothenburg, 40530 Gothenburg, Sweden; 5Department of Microbiology and Immunology, Institute of Biomedicine, Sahlgrenska Academy, University of Gothenburg, 40530 Gothenburg, Sweden; 6Department of Infectious Diseases, Sahlgrenska University Hospital, 41685 Gothenburg, Sweden; vanda.friman@gu.se (V.F.); magnus.gisslen@infect.gu.se (M.G.); 7Department of Infectious Diseases, Institute of Biomedicine, Sahlgrenska Academy, University of Gothenburg, 40530 Gothenburg, Sweden; 8Department of Molecular and Clinical Medicine, Institute of Medicine, Sahlgrenska Academy, University of Gothenburg, 40530 Gothenburg, Sweden; 9Department of Nephrology, Skåne University Hospital, 21428 Malmö, Sweden; carin.wallquist@skane.se

**Keywords:** COVID-19, solid organ transplant recipients, transplantation, immunosuppression, vaccines

## Abstract

Increased COVID-19-related morbidity and mortality have been reported in solid organ transplant recipients (SOTRs). Most studies are underpowered for rigorous matching. We report infections, hospitalization, ICU care, mortality from COVID-19, and pertinent vaccination data in Swedish SOTRs 2020–2021. We conducted a nationwide cohort study, encompassing all Swedish residents. SOTRs were identified with ICD-10 codes and immunosuppressant prescriptions. Comparison cohorts were weighted based on a propensity score built from potential confounders (age, sex, comorbidities, socioeconomic factors, and geography), which achieved a good balance between SOTRs and non-SOTR groups. We included 10,372,033 individuals, including 9073 SOTRs. Of the SARS-CoV-2 infected, 47.3% of SOTRs and 19% of weighted comparator individuals were hospitalized. ICU care was given to 8% of infected SOTRs and 2% of weighted comparators. The case fatality rate was 7.7% in SOTRs, 6.2% in the weighted comparison cohort, and 1.3% in the unweighted comparison cohort. SOTRs had an increased risk of contracting COVID-19 (HR = 1.15 *p* < 0.001), being hospitalized (HR = 2.89 *p* < 0.001), receiving ICU care (HR = 4.59 *p* < 0.001), and dying (HR = 1.42 *p* < 0.001). SOTRs had much higher morbidity and mortality than the general population during 2020–2021. Also compared with weighted comparators, SOTRs had an increased risk of contracting COVID-19, being hospitalized, receiving ICU care, and dying. In Sweden, SOTRs were vaccinated earlier than weighted comparators. Lung transplant recipients had the worst outcomes. Excess mortality among SOTRs was concentrated in the second half of 2021.

## 1. Introduction

During the early phases of the COVID-19 pandemic, solid organ transplant recipients (SOTRs) were reported to have high morbidity and mortality. Initially, 75% of SOTRs diagnosed with COVID-19 required hospital admission. Of these, nearly 40% required intensive care. Mortality among hospitalized patients in large cohorts of SOTRs with COVID-19 has ranged from 10% to 20% [1,2,3,4,5,6].

Although these outcomes are concerning, it has yet to be determined how they differ from properly matched controls since it is challenging to match enough relevant parameters due to limitations in the available datasets. In meta-analyses, mortality has been estimated to range from 1.1 to 1.5 times that of matched controls [7,8]. For SOTRs, such as the general population, the preponderance of evidence supports that age and comorbidities, rather than immunosuppression-related factors, drive COVID-19 mortality [3,9,10]. While some studies have indicated that lung transplant recipients may fare worse than other organ recipients [11,12,13], most comparative studies have been underpowered to answer questions concerning the influence of organ transplant type [8].

Due to early publications showing high morbidity and mortality amongst SOTRs, this patient group was prioritized in many countries to receive vaccinations early [14]. This approach aimed to reduce the disease burden for the group despite the paucity of vaccine data on immunocompromised patients. Later vaccine studies on SOTRs have shown a much lower seroconversion rate than in the general population [15]. The effect of early prioritization in vaccine programs has not been studied in large, well-matched cohorts.

Our study leverages the data from the Swedish COVID-19 Investigation for Future Insights—a Population Epidemiology Approach using Register Linkage (SCIFI-PEARL) project [16]. We identified almost all SOTRs in Sweden and used the rest of the Swedish population as a comparison cohort (non-SOTR group). In this study, we used a complete national dataset to study the relative and absolute risk of infection, hospitalization, ICU care, and mortality in SOTRs overall and organ subgroups, compared with weighted comparison cohorts.

## 2. Materials and Methods

### 2.1. Study Design

We conducted a cohort study to examine the risk of SARS-CoV-2 infection and severe outcomes from COVID-19 in SOTRs. We included all individuals who were alive and living in Sweden on 1 January 2020 (*n* = 10,372,033) and categorized them into SOTR and non-SOTR groups according to their transplantation status at the baseline (defined in detail below). The SOTR group was further divided into four transplanted-organ subgroups (kidney, liver, heart, and lung), and analyses were conducted for SOTR overall and for each transplanted-organ subgroup. In each analysis, all individuals were followed until they developed an outcome event or were censored. Censoring occurred at emigration, death, or the end of follow-up (31 December 2021).

### 2.2. Data Sources

The data were extracted from the pseudonymized database of the SCIFI-PEARL project [16]. These data originated from the registers in Table 1, with individuals linked between registers through the Swedish personal identification numbers [17].

### 2.3. Study Variables

#### 2.3.1. Transplantation Status

To identify SOTRs, we used International Classification of Disease revision 10 (ICD-10) codes and Nordic Medico-Statistical Committee (NOMESCO) Classification of Surgical Procedures (NCSP) codes (Appendix B, Table A1) registered from 1 January 2015 to 31 December 2019 for transplant status and transplant operations to identify potential SOTRs. Four types of SOTRs were identified: kidney (KTx), heart (HTx), lung (LuTx), and liver (LTx). Recipients of other organs, e.g., pancreas, uterus, hand, and small intestine, were not included because their numbers were too low for a meaningful analysis. Combined transplant recipients were categorized in the following order of priority: lung, heart, kidney, and liver. This was based on the assumptions that lung recipients may be uniquely susceptible to the effects of COVID-19, thoracic organ recipients generally receive more immunosuppression than abdominal organ recipients in Sweden, and liver recipients receive the least.

Immunosuppressant drugs prescribed to patients during 2019 were identified from the NPDR and included tacrolimus, ciclosporin, mycophenolic acid, prednisolone, sirolimus, everolimus, methotrexate, and azathioprine (Appendix B, Table A2). Individuals not exposed to any of the listed immunosuppressants were excluded from the SOTR group. Patients with only ICD-10 status codes (Z-codes) went through an additional step of cleaning. Because transplant status codes are registered each time a patient has a hospital visit, it is likely that patients could occasionally be coded incorrectly. The frequency of each status code was counted, and its proportion related to the total count of all status codes was calculated. The status codes were considered true in the absence of a NCSP code, indicating that the patient underwent transplantation prior to 2015, only if the frequency was greater than five. If multiple different status codes were found, implying more than one type of transplant, in the absence of confirmatory NCSP codes, they were only considered true if the proportion of the least frequent code was greater than 20% of all transplant status codes for a given individual.

Uniquely for KTx recipients, we had to identify patients who had experienced graft failure and were back on dialysis as non-transplanted. This was to avoid biasing the results in the KTx group with those from chronic dialysis patients who may have other vulnerabilities [25]. Dialysis patients were identified by using specific ICD-10 and NSCP codes (Appendix B, Table A1). KTx recipients who had dialysis records later than 40 days after their kidney transplantation procedure were excluded from the KTx group but could be retained in another transplantation group in the event of a combined transplant. The rest of the population of Sweden (non-SOTR) was considered the comparative cohort.

#### 2.3.2. COVID-19 Outcomes

Outcomes were defined as seen in Table 2:

#### 2.3.3. Covariates

Age, sex, comorbidities (Charlson comorbidity index (CCI) according to Quan’s ICD-10 CCI algorithm [26,27]), education level (primary, secondary, tertiary, unknown), income (quartiles of disposable income plus unknown), country of birth (Sweden or immigrant from low income, lower-middle income, upper-middle income, high-income countries as defined by the World Bank), marital status (married, not married (including widow[er]s)), and healthcare region (six medical care regions in Sweden).

### 2.4. Statistical Analyses

Baseline characteristics are presented as frequencies and percentages for categorical variables and as mean values and standard deviation (SD) for continuous variables.

The balance between SOTR and non-SOTR groups was achieved by propensity score weighting (PSW) to minimize confounding effects. The propensity score (ps) for each individual to be in the SOTR group was calculated using a logistic model, where all the covariates listed above were included. Balance was achieved by aiming at estimating the average treatment effects in the treated (ATT), and therefore all patients in the SOTR group were designated the weight as 1.0, while the non-SOTR group subjects were assigned a weight of ps/(1 − ps). The balance between the groups was evaluated by inspecting the standardized mean difference (SMD) of all covariates for the individuals with a valid ps value. For subgroup analyses, we used the same approach in each subgroup, thus creating appropriately balanced comparative cohorts for each organ recipient type (Appendix B, Table A3).

Positive vaccination status was defined as having received two doses of vaccine. Population quartiles for the date of positive vaccination were calculated, respectively, for each group considering the weights.

To illustrate the course of the pandemic in Sweden, the daily number of infections and prevalence of different SARS-CoV-2 variants of concern (VOC) were calculated from the full population data and plotted over time. The VOCs were defined according to national data from the SmiNet from genome sequencing of selected samples over time, and the prevalence was smoothed using a 7-day moving average.

Survival analysis was used to assess all outcomes, where 1 January 2020, was set as the day of entering the cohort and exiting the cohort as the earliest outcome, death, emigration, or end of follow-up. The survival curves of the groups are illustrated using the weighted Kaplan-Meier failure curve. Hazard ratios (HR) and 95% confidence intervals (95% CI) for SOTRs compared with non-SOTRs were calculated using the weighted Cox proportional-hazard models.

Cox regression analyses of COVID-19-related deaths before and after the 50th percentile of SOTRs had received a second dose of the vaccine were conducted.

All analyses were performed using STATA (version 17) and R (version 4.0.2).

The study follows the STROBE and RECORD guidelines, and checklists can be found in the Appendix A.

## 3. Results

### 3.1. Patient Characteristics

We identified 9073 SOTRs: 5967 KTx, 1866 LTx, 778 HTx, and 463 LuTx (Table 3). Compared to non-SOTRs (unweighted comparison cohort, n = 10,362,960), SOTRs were older, more likely to be male, had more comorbidities, and demonstrated some differences in socioeconomic and geographic variables. After weighting, the SOTR group and the balanced comparison cohort were comparable in their characteristics (Table 4). The same weighting procedure resulted in comparable characteristics for subgroups (Appendix B, Table A3).

### 3.2. Outcomes

#### 3.2.1. Comparisons with the General Population:

During the period 1 January 2020 to 31 December 2021, 12.5% (n = 1,294,658) of the Swedish general population (i.e., unweighted comparison cohort, n = 10,362,960) were diagnosed with COVID-19, 0.72% (n = 74,245) were hospitalized, ICU care was given to 0.08% (n = 8101), and COVID-19 was a cause of death in 0.16% (n = 16,582). Of the total SOTR population (n = 9073), 14.6% (n = 1328) received a diagnosis of COVID-19, 6.92% (n = 628) were hospitalized, ICU care was given to 1.17% (n = 106), and COVID-19 was a cause of death in 1.12% (n = 102). Thus, compared with the general population, infection was 1.17-fold more common, hospitalization was 9.6-fold more common, ICU care was 15-fold more common, and death due to COVID-19 was 7-fold more common in the SOTR population. However, the SOTR population was older, had a higher percentage of males, and had more comorbidities than the general population.

#### 3.2.2. Comparisons with Weighted Comparison Cohorts

Hospitalization was necessary in 47.3% of the 1328 SOTRs and 19% of the weighted comparison group with COVID-19. ICU care was given to 8% of SOTRs and 2% of weighted comparators with COVID-19. The case fatality rate (CFR) for 2020–2021 was 7.7% in SOTRs, 6.2% in weighted comparators (n = 102/1328 vs. 71/1151), and 1.3% in the unweighted comparison cohort (n = 16,582/10,362,960). The CFR was 6.7% for KTx, 8.3% for LTx, 10.4% for HTx, and 15.4% for LuTx (Table 5).

There was considerable variation in outcomes between organ subgroups (Table 6). Only KTx recipients were diagnosed with COVID-19 significantly more often than the weighted comparators. Hospitalization was more common in all SOTR subgroups. While ICU admission was more common in SOTRs overall, there were some outliers in the subgroups. LuTx recipients were far more likely to require ICU care, whereas HTx recipients were not significantly different than the weighted comparators. 30-day mortality was only significantly higher for LuTx recipients, whereas 60-day mortality and COVID-related death were increased for LuTx-, HTx-, and KTx-recipients, but not LTx recipients.

The median time from infection to hospital admission was the same in SOTRs and the weighted comparison cohort. The median days to ICU admission was longer in the SOTR group, 7 days vs. 4 days. SOTRs had a longer median time from diagnosis to COVID-19-related death than the comparison cohort: 16 days vs. 10 days, respectively. The subgroup variation was considerable with 19 in KTx, 15 in LTx, 8 in HTx, and 20 days in LuTx, while all four weighted comparison subgroups had a median of 10 days (Table 7).

#### 3.2.3. Variations over Time throughout the Pandemic

The pandemic began to accelerate in Sweden in March 2020 (Figure 1). Infection curves for SOTRs and the weighted comparison cohort followed each other closely for the whole study period, while hospital and ICU admission curves diverged strongly (Figure 2).

Mortality curves for SOTRs and the weighted comparison cohort followed each other closely for the first and second waves of SARS-CoV-2 infections. In April 2021, at the peak of the third wave, when the Alpha VOC was dominant in Sweden, they diverged for the first time.

During the second wave, the failure curves of both groups started to rise in November 2020. The rise in the weighted comparison cohort ceased and went back to a flatter curve in February 2021, but such a turning point happened later in the SOTR group, in June 2021 (Figure 1).

#### 3.2.4. SARS-CoV-2 Vaccination

SOTRs were vaccinated earlier than the comparison cohort. The first, second, and third quartiles of SOTRs received their second dose between April and May 2021. The first, second, and third quartiles of the weighted comparison cohorts received their second dose in May, June, and September 2021, respectively (Appendix B, Table A4). When examining vaccination times for organ subgroups, the transplant groups were vaccinated at similar times, but the third quartile of weighted comparators for HTx and LTx recipients were vaccinated later than weighted comparators for LuTx and KTx recipients.

Cox regression analyses of COVID-19-related death before and after 21 April 2021, the date at which the 50th percentile of SOTRs had received a second dose, showed no difference before this timepoint (HR = 1.07, 95% CI 0.84–1.35, *p* = 0.59), but a large difference thereafter (HR = 5.04, 95% CI 3.42–7.24, *p* < 0.001), compared to controls.

## 4. Discussion

Using multiple registries, we conducted a comprehensive analysis of COVID-19-related morbidity and mortality in SOTRs as compared with comparison cohorts balanced for age, sex, comorbidities, and socioeconomic factors using propensity score weighting, utilizing the entire population of Sweden. We observed that SOTRs were diagnosed with COVID-19 more often than weighted comparators and had considerably increased associated morbidity, as demonstrated by significantly higher rates of hospital admissions and ICU care. Mortality was also significantly increased and delayed compared with weighted comparators; the CFR for COVID-19 in SOTRs was 7.7%, compared with 6.2% in weighted comparators and 1.3% in the general population (unweighted comparators). LuTx recipients had the highest burden of morbidity and mortality.

SOTRs were more likely to receive a positive test result or a diagnosis of COVID-19 compared to their weighted comparators. This is surprising since SOTRs, unlike the general population, were recommended to self-isolate [28]. This finding, also shown in another study [29], could be explained in several ways: increased risk of exposure from frequent healthcare contacts, greater susceptibility to a lower threshold dose of SARS-CoV-2 for infection; greater likelihood of symptomatic rather than asymptomatic infection; or ascertainment bias due to increased alertness and testing in this population. While these may all be true, it is evident that SOTRs also have a 3-fold increased risk of hospitalization, suggesting that they indeed suffer more severe COVID-19 than weighted comparators.

The metric of hospital admissions may be perceived as subjective since it could be affected by bias, with a higher willingness to admit a transplant patient for observation than a non-transplant patient. However, ICU admissions during the pandemic were based on very strict and objective criteria, and SOTRs were 4.6-fold more likely to be admitted to the ICU than weighted comparators. While all organ subgroups were at increased risk of requiring ICU care, there was a considerable variation, with LuTx recipients being almost 14-fold more likely to be admitted to the ICU than weighted comparators.

The difference in mortality was less pronounced, with 30-day all-cause mortality being 20% higher than weighted comparators and total mortality with COVID-19 as a cause of death being 42% higher. Similar to another recent study, we observed that the disease course was often prolonged in SOTRs, leading to a high proportion of the excess mortality occurring after the 30-day mortality window [11].

When dividing the analysis of COVID-19-related death into periods before and after the 50th percentile of SOTRs had received a second vaccine dose (April 2021), we found no difference in the HR for COVID-19-related death compared to the weighted comparators before this date and a large difference thereafter. Surprisingly, mortality was similar in the overall SOTR group and the weighted comparison cohort until April 2021. At this point, the curves diverged, and SOTRs had worse survival. One could interpret this divergence as a continuation of the upward slope of the SOTR mortality curve, while the weighted comparison cohort has an earlier break in the curve.

In Sweden, the first COVID-19 vaccines to be approved were Comirnaty (BioNTech/Pfizer) on 21 December 2020, Spikevax (Moderna) on 6 January 2021, and Vaxzevria (AstraZeneca) on 20 January 2021. Vaccinations commenced 27 December 2020 and were initially prioritized according to age and later according to certain medical conditions. SOTRs were prioritized for vaccination in March 2021 with the mRNA vaccines Comirnaty and Spikevax. The third quartile of SOTRs received a second dose before the first quartile of weighted comparators received a second dose. Thus, it was surprising that mortality curves for COVID-19-related deaths started diverging in the direction observed in April 2021.

One possible explanation is that the divergence reflects differences in vaccine efficacy. Transplant patients were comparatively poorly protected from two doses of vaccines [15], but this was not common knowledge at the time since SOTRs were excluded from the initial vaccine trials [30,31]. Multiple studies have later shown that SOTRs respond poorly to SARS-CoV-2 vaccines, with a recent meta-analysis showing 9.5% seroconversion after one dose, 44% after two doses, and 55% after three doses [32]. In contrast, seroconversion following a single dose of vaccine was common in the general population [33].

After more than a year of self-isolation, many newly vaccinated people may have felt that they could resume a more normal lifestyle [34]. Vaccinations may have caused a false sense of security amongst SOTRs and consequently an increase in severe cases amongst SOTRs, but not in the comparatively vaccine-protected weighted comparators. Since vaccination was largely carried out at non-specialized centers, it is unlikely that specific caveats were provided to patients concerning continued self-isolation after two doses, despite previous vaccine studies, e.g., influenza, showing that SOTRs may respond less well to vaccination [35]. Government guidelines also discouraged serological testing, impeding the possibility of informing non-responders of their potential continued risk.

The observed delay from infection to death in SOTRs, except for HTx recipients, may have also caused a slight right-shift in the SOTR mortality curve. One possible explanation for this delay is that SOTRs may have been more likely to have a reinfection with a poor outcome. While a previous study has shown that around 90% of transplant recipients seroconvert after contracting COVID-19, a minority do not [36]. Reinfections within one year were not initially counted as a new event in the SmiNet. It is therefore, theoretically, possible that some reinfections would be misclassified as a continuation of the initial case. However, we do not believe this has much influence on our results since the time from infection to hospitalization is similar in SOTRs and the weighted comparison cohort. Another explanation for the delay from infection to death is that COVID-19 may start a protracted inflammatory process in the transplanted organ, particularly in LuTX recipients, who were found to have the longest time from infection to death in our study. Other contributing factors could be slower viral clearance, more venous thromboembolic events, and more frequent secondary bacterial infections. The rapid transition from infection to death with low ICU admission rates in the HTx group may point to a different and more rapidly evolving mechanism.

In most publications, only 30-day mortality is used as an endpoint. This has been criticized in a recent study, which also showed delayed mortality in SOTRs [11]. On one hand, 30-day mortality is a standard metric that allows easy and meaningful comparisons between regions. On the other hand, the choice of this metric may not be due to its inherent qualities but rather to this data being more readily available than cause-of-death data. Since death from COVID-19 is delayed in the SOTR group, comparative studies using 30-day mortality could risk underestimating mortality among SOTRs.

This study has several strengths. It offers a comprehensive analysis covering 9073 of approximately 10,000 SOTRs living in Sweden in 2020. Part of the shortfall could be made up of recipients for the 724 organs that were transplanted during that year [37]. The comparison cohort comprised the full population of Sweden, over 10.3 million individuals, supporting balancing on a large number of factors using state-of-the-art propensity score weighting methods. Previous studies have struggled to match adequately due to relatively small groups in case series, a lack of potential matches, and relevant matching data in epidemiological studies [8].

Sweden is uniquely positioned to provide epidemiological data since it has multiple high-quality national registries, and every person in the country has a national identification number, making them traceable across registries [17]. Unlike most other countries, data in Swedish registries are nearly complete and rarely suffers from loss of follow-up or patients opting out since normally only patients moving out of the country would be lost. However, income and education data for children and some young adults will be listed as unknown. Sweden has a single-payer healthcare system, allowing free access for all citizens. While this may reduce the influence of socioeconomic factors on outcomes, it does not remove them entirely [38]. It has been shown that socioeconomic status, minority background, and regional differences influence the risk of exposure and outcomes [39]. We believe this is the largest published series of COVID-19 in SOTRs, with the whole country’s population as the comparator. In addition, this study includes complete vaccination data on the study population as well as comprehensive pharmaceutical prescription information.

There are also some weaknesses inherent to the study design. In a registry study with base data from over 10 million individuals, care must be taken not to erroneously include the wrong patients in the study group. Due to a stringent definition of SOTRs requiring both the presence of ICD-10 codes, prescribed immunosuppressants, and being off dialysis, the total transplant population is marginally below the expected value. Furthermore, separate analyses of combined organ recipients and uncommon transplant types are lacking, and patients who underwent transplantation during the study period were not included in the SOTR group. We also did not have access to body mass index or ethnicity for balancing since there are no official Swedish statistics on ethnicity. However, data are available on country of birth, and since high-volume non-white immigration to Sweden is a relatively modern phenomenon, using country of origin partially ameliorates this shortcoming.

A limitation of many comparative studies is that they are often heavily skewed towards hospitalized patients [8], thereby any differences in the denominator of patients infected who could potentially require hospital admission are not considered. By including all Swedish residents in the study and identifying those with a positive test or diagnosis, we aimed to elucidate any potential differences in progression to severe outcomes between groups. We observed a higher rate of infections in the SOTR group. While this may be due to the increased frequency of testing, which may bias towards underestimating the severe outcome measures, the fact that this group progressed to requiring hospitalization and ICU care more often than weighted comparators throughout the pandemic strengthens our conclusion that SOTRs are indeed affected by higher COVID-19-related morbidity.

Our findings have several implications. They validate the general pandemic guidelines for SOTRs to demonstrate increased care and vigilance when faced with a new pathogen. In COVID-19 studies on SOTRs, the 30-day mortality metric may be inadequate, and longer follow-up is necessary. Furthermore, since excess mortality was concentrated in the latter half of 2021, it may be prudent, in future pandemics, to adopt a conservative approach to post-vaccination self-isolation guidelines for vulnerable groups such as SOTRs until the evidence supports liberalizing them. SARS-CoV-2 vaccines appear to have lacked effectiveness in the Swedish SOTR population during the study period. This study also serves as a cautionary note for extrapolating vaccine immunogenicity and efficacy data from the general population to immunosuppressed groups before the availability of evidence to support it. It also suggests that pharmaceutical companies should prioritize transplant patients in Phase III trials on vaccines, as the experience from this pandemic has shown that they have the worst response of all immunosuppressed groups [15]. In the absence of such prioritization, serological follow-up should be mandatory in clinical practice when using new vaccines in SOTRs prior to sufficient evidence of efficacy. Implementation of improved vaccine protocols, close monitoring, and better-informed post-vaccination guidelines may be crucial to better protecting susceptible populations, such as SOTRs, in other possible future pandemics.

## 5. Conclusions

SOTRs had much higher morbidity and mortality than the general population during 2020–2021. Also compared with weighted comparators, SOTRs had an increased risk of contracting COVID-19, being hospitalized, receiving ICU care, and dying. In Sweden, SOTRs were vaccinated earlier than weighted comparators. Lung transplant recipients had the worst outcomes. Excess mortality among SOTRs was concentrated in the second half of 2021.

## Figures and Tables

**Figure 1 viruses-16-00271-f001:**
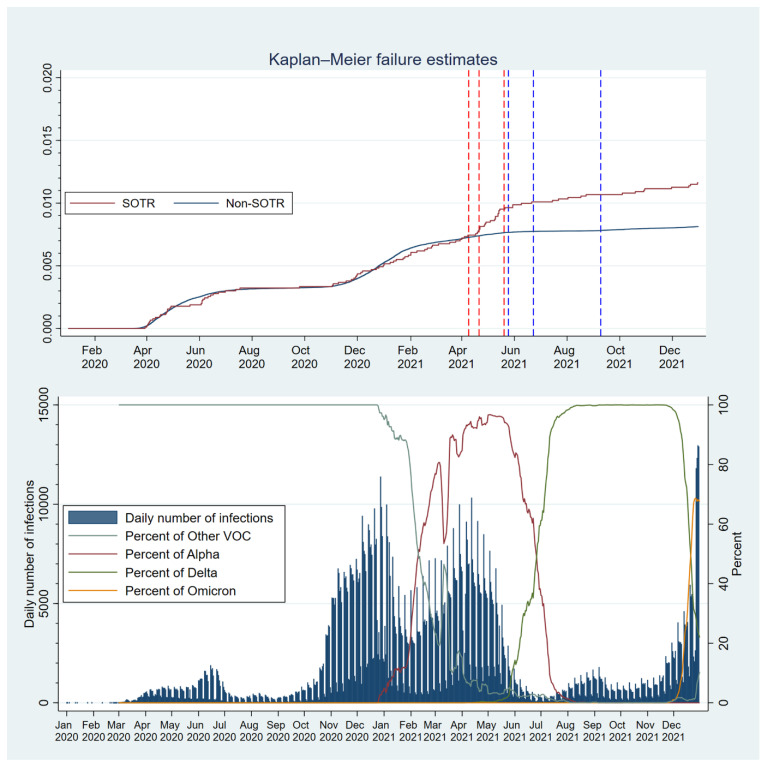
Weighted Kaplan-Meier failure estimates (COVID-19-related death) for solid organ transplant recipients (SOTR) and a non-SOTR weighted comparison cohort. Red vertical dotted lines show when the first, second, and third quartiles of the SOTR cohort had received two doses of the SARS-CoV-2 vaccine. Blue vertical dotted lines show when the first, second, and third quartiles of the weighted comparison cohort had received two doses of vaccine. There is no line for the fourth quartile in either group since vaccine coverage never reached 100%. The timeline of the daily number of infections and percentage of variants of concern (VOC) in sequenced samples in Sweden is shown in the lower part with the same time axis.

**Figure 2 viruses-16-00271-f002:**
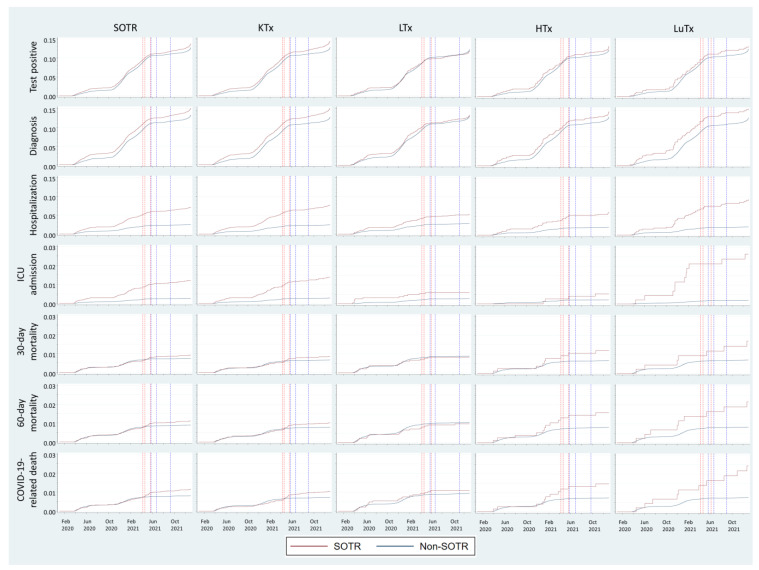
Weighted Kaplan-Meier failure estimates (test positive, infection, hospitalization, ICU admission, 30-day mortality, 60-day mortality, COVID-19-related death) for SOTRs and non-SOTR weighted comparison cohorts. Red vertical dotted lines show when the first, second, and third quartiles of the SOTR cohorts had received two doses of the SARS-CoV-2 vaccine. Blue vertical dotted lines show when the first, second, and third quartiles of the weighted comparison cohorts had received two doses of vaccine. There is no line for the fourth quartile in any group since vaccine coverage never reached 100%. SOTR: Solid organ transplant recipient KTx: Kidney Transplanted, LTx: Liver Transplanted, HTx: Heart Transplanted, LuTx: Lung Transplanted.

**Table 1 viruses-16-00271-t001:** Data sources and retrieved variables.

Register	Retrieved Variables
National Patient Register (NPR) [18]	Transplant status (exposure) Selected comorbidities recorded in inpatient care or during consultations in specialist ambulatory care (potential confounders) COVID-19-related hospitalization (outcome)
Swedish Intensive Care Register (SIR) [19]	COVID-19-related ICU records (outcome)
National Prescribed Drug Register (NPDR) [20]	Dispensed prescriptions of immunosuppressive drugs (exposure)
National database of notifiable diseases (SmiNet) [21]	Positive SARS-CoV-2 polymerase chain reaction (PCR) test results (outcome)
Cause-of-death register (CoDR) [22]	Date and causes of death (outcome)
National Vaccine Registry (NVR) [23]	Date and type of COVID-19 vaccination (potential confounders)
Longitudinal integrated database for health insurance and labor market studies (LISA) [24]	Demographic and socioeconomic characteristics (potential confounders)

**Table 2 viruses-16-00271-t002:** Description of outcome variables.

COVID-19 Outcome	Description
Test-positive	PCR test positive for SARS-CoV-2 from SmiNet
Infection	The earliest of the following: (1) test-positive; (2) receiving a diagnosis of COVID-19 infection (ICD-10 U07.1, U07.2) as a primary or secondary diagnosis from the NPR; (3) death due to COVID-19 (COVID-19-related death) from the CoDR
Hospitalization	Inpatient records with a primary or secondary diagnosis of COVID-19 from the NPR
ICU admission	Intensive care records with a primary or secondary diagnosis of COVID-19 from the SIR
30-day mortality (all-cause)	Death (of any cause) within 30 days from a test-positive
60-day mortality (all-cause)	Death (of any cause) within 60 days from a test-positive
COVID-19-related death	COVID-19 diagnosis as an underlying or contributing cause of death from the CoDR
Case fatality rate (CFR)	Calculated using COVID-19-related death as the numerator and infection as the denominator (as defined above)

**Table 3 viruses-16-00271-t003:** Number of solid organ transplant recipients (SOTRs) in 2015–2019 with different types of transplantation, characteristics at baseline 1 January 2020, and with various types of immunosuppressive drugs in 2019, in Sweden.

	Overall SOTRs	KTx	LTx	HTx	LuTx
**Count**	9073	5967	1866	778	462
**Age, mean(SD)**	54.90 (16.33)	55.35 (15.54)	54.11 (18.31)	52.58 (17.98)	56.21 (14.37)
**Gender (female), *n* (%)**	3342 (36.8%)	2166 (36.3%)	700 (37.5%)	235 (30.2%)	241 (52.2%)
**Education level, *n* (%)**					
Primary	1784 (19.7%)	1187 (19.9%)	357 (19.1%)	140 (18.0%)	100 (21.6%)
Secondary	4128 (45.5%)	2769 (46.4%)	798 (42.8%)	359 (46.1%)	202 (43.7%)
Tertiary	2801 (30.9%)	1832 (30.7%)	588 (31.5%)	227 (29.2%)	154 (33.3%)
Unknown	360 (4.0%)	179 (3.0%)	123 (6.6%)	52 (6.7%)	6 (1.3%)
**Charlson comorbidity index, median, *n* (%)**	2	1	3	2	1
0	305 (3.4%)	0 (0.0%)	0 (0.0%)	198 (25.4%)	107 (23.2%)
1	3850 (42.4%)	3617 (60.6%)	0 (0.0%)	71 (9.1%)	162 (35.1%)
2	2201 (24.3%)	1089 (18.3%)	771 (41.3%)	249 (32.0%)	92 (19.9%)
3	1092 (12.0%)	704 (11.8%)	193 (10.3%)	146 (18.8%)	49 (10.6%)
4	858 (9.5%)	314 (5.3%)	468 (25.1%)	46 (5.9%)	30 (6.5%)
5	330 (3.6%)	116 (1.9%)	165 (8.8%)	41 (5.3%)	8 (1.7%)
6	204 (2.2%)	45 (0.8%)	143 (7.7%)	9 (1.2%)	7 (1.5%)
7	127 (1.4%)	55 (0.9%)	54 (2.9%)	12 (1.5%)	6 (1.3%)
8	65 (0.7%)	15 (0.3%)	48 (2.6%)	2 (0.3%)	0 (0.0%)
9	22 (0.2%)	8 (0.1%)	11 (0.6%)	3 (0.4%)	0 (0.0%)
10	11 (0.1%)	3 (0.1%)	7 (0.4%)	1 (0.1%)	0 (0.0%)
11	6 (0.1%)	1 (0.0%)	4 (0.2%)	0 (0.0%)	1 (0.2%)
12	2 (0.0%)	0 (0.0%)	2 (0.1%)	0 (0.0%)	0 (0.0%)
**Income, *n* (%)**					
Low	2098 (23.1%)	1404 (23.5%)	460 (24.7%)	134 (17.2%)	100 (21.6%)
Low-Middle	2281 (25.1%)	1483 (24.9%)	431 (23.1%)	221 (28.4%)	146 (31.6%)
Middle-High	2212 (24.4%)	1480 (24.8%)	430 (23.0%)	201 (25.8%)	101 (21.9%)
High	2274 (25.1%)	1516 (25.4%)	459 (24.6%)	187 (24.0%)	112 (24.2%)
Unknown	208 (2.3%)	84 (1.4%)	86 (4.6%)	35 (4.5%)	3 (0.6%)
**Country of birth, *n* (%)**					
Sweden	7324 (80.7%)	4780 (80.1%)	1470 (78.8%)	676 (86.9%)	398 (86.1%)
Low-income	522 (5.8%)	324 (5.4%)	140 (7.5%)	29 (3.7%)	29 (6.3%)
Low-Middle-income	291 (3.2%)	215 (3.6%)	62 (3.3%)	8 (1.0%)	6 (1.3%)
Middle-High-income	263 (2.9%)	188 (3.2%)	49 (2.6%)	22 (2.8%)	4 (0.9%)
High-income	548 (6.0%)	377 (6.3%)	117 (6.3%)	33 (4.2%)	21 (4.5%)
Unknown	125 (1.4%)	83 (1.4%)	28 (1.5%)	10 (1.3%)	4 (0.9%)
**Marital status (not married *), *n* (%)**	4866 (53.6%)	3200 (53.6%)	1008 (54.0%)	403 (51.8%)	255 (55.2%)
**Healthcare region, *n* (%)**					
North	894 (9.9%)	550 (9.2%)	201 (10.8%)	106 (13.6%)	37 (8.0%)
Stockholm	1920 (21.2%)	1241 (20.8%)	453 (24.3%)	122 (15.7%)	104 (22.5%)
Southeast	909 (10.0%)	614 (10.3%)	176 (9.4%)	78 (10.0%)	41 (8.9%)
South	1582 (17.4%)	1041 (17.4%)	299 (16.0%)	141 (18.1%)	101 (21.9%)
Uppsala-Örebrö	1917 (21.1%)	1405 (23.5%)	303 (16.2%)	127 (16.3%)	82 (17.7%)
West	1851 (20.4%)	1116 (18.7%)	434 (23.3%)	204 (26.2%)	97 (21.0%)
**Immunosuppression**					
Ciclosporin	1530 (16.9%)	935 (15.7%)	139 (7.4%)	202 (26.0%)	254 (54.9%)
Tacrolimus	7162 (78.9%)	4774 (80.0%)	1625 (87.1%)	554 (71.2%)	209 (45.1%)
Methotrexate	6 (0.1%)	1 (0.0%)	2 (0.1%)	1 (0.1%)	2 (0.4%)
Prednisolone	6876 (75.8%)	5102 (85.5%)	920 (49.3%)	399 (51.3%)	455 (98.3%)
Sirolimus	63 (0.7%)	35 (0.6%)	22 (1.2%)	2 (0.3%)	4 (0.9%)
Everolimus	806 (8.9%)	294 (4.9%)	123 (6.6%)	229 (29.4%)	160 (34.6%)
Mycophenolic acid	6353 (70.0%)	4457 (74.7%)	827 (44.3%)	671 (86.2%)	398 (86.0%)
Azathioprine	662 (7.3%)	394 (6.6%)	179 (9.6%)	53 (6.8%)	36 (7.8%)
Average number of prescribed immunosuppressants per patient	2.59	2.68	2.06	2.71	3.28

KTx: Kidney Transplanted, LTx: Liver Transplanted, HTx: Heart Transplanted, LuTx: Lung Transplanted. * Not married includes widow[er]s.

**Table 4 viruses-16-00271-t004:** Characteristics of the solid organ transplant recipients (SOTR) and the non-SOTR comparison cohort before and after ATT weighting on a propensity score.

	Before Weighting			After Weighting		
	Non-SOTR	SOTR	SMD	Non-SOTR	SOTR	SMD
**Count (or equivalent weighted count)**	10,362,960	9073		9070.49	9073	
**Age, mean (SD)**	40.74 (23.98)	54.90 (16.33)	0.690	54.88 (21.86)	54.90 (16.33)	0.001
**Gender (female), *n* (%)**	5,148,823 (49.7%)	3342 (36.8%)	0.262	3341.1 (36.8%)	3342.0 (36.8%)	<0.001
**Education level, *n* (%)**			0.549			0.001
Primary	1,623,103 (15.7%)	1784 (19.7%)		1785.4 (19.7%)	1784.0 (19.7%)	
Secondary	3,511,577 (33.9%)	4128 (45.5%)		4128.1 (45.5%)	4128.0 (45.5%)	
Tertiary	3,031,146 (29.2%)	2801 (30.9%)		2797.1 (30.8%)	2801.0 (30.9%)	
Unknown	2,197,134 (21.2%)	360 (4.0%)		359.9 (4.0%)	360.0 (4.0%)	
**Charlson comorbidity index, median, *n* (%)**	0	2	3.490	2	2	0.003
0	9,301,653 (89.8%)	305 (3.4%)		305.0 (3.4%)	305.0 (3.4%)	
1	453,394 (4.4%)	3850 (42.4%)		3856.6 (42.5%)	3850.0 (42.4%)	
2	413,578 (4.0%)	2201 (24.3%)		2201.7 (24.3%)	2201.0 (24.3%)	
3	74,577 (0.7%)	1092 (12.0%)		1086.4 (12.0%)	1092.0 (12.0%)	
4	45,322 (0.4%)	858 (9.5%)		857.7 (9.5%)	858.0 (9.5%)	
5	13,808 (0.1%)	330 (3.6%)		326.7 (3.6%)	330.0 (3.6%)	
6	46,807 (0.5%)	204 (2.2%)		204.1 (2.2%)	204.0 (2.2%)	
7	7322 (0.1%)	127 (1.4%)		126.7 (1.4%)	127.0 (1.4%)	
8	4196 (0.0%)	65 (0.7%)		64.8 (0.7%)	65.0 (0.7%)	
9	1431 (0.0%)	22 (0.2%)		21.9 (0.2%)	22.0 (0.2%)	
10	646 (0.0%)	11 (0.1%)		11.0 (0.1%)	11.0 (0.1%)	
11	177 (0.0%)	6 (0.1%)		5.9 (0.1%)	6.0 (0.1%)	
12	49 (0.0%)	2 (0.0%)		2.0 (0.0%)	2.0 (0.0%)	
**Income, *n* (%)**			0.534			<0.001
Low	1,989,214 (19.2%)	2098 (23.1%)		2097.9 (23.1%)	2098.0 (23.1%)	
Low-Middle	2,133,870 (20.6%)	2281 (25.1%)		2280.0 (25.1%)	2281.0 (25.1%)	
Middle-High	2,199,276 (21.2%)	2212 (24.4%)		2211.8 (24.4%)	2212.0 (24.4%)	
High	2,198,569 (21.2%)	2274 (25.1%)		2272.9 (25.1%)	2274.0 (25.1%)	
Unknown	1,842,031 (17.8%)	208 (2.3%)		207.9 (2.3%)	208.0 (2.3%)	
**Country of birth, *n* (%)**			0.048			0.001
Sweden	8,304,260 (80.1%)	7324 (80.7%)		7319.9 (80.7%)	7324.0 (80.7%)	
Low-income	623,055 (6.0%)	522 (5.8%)		522.2 (5.8%)	522.0 (5.8%)	
Low-Middle-income	414,380 (4.0%)	291 (3.2%)		291.9 (3.2%)	291.0 (3.2%)	
Middle-High-income	300,270 (2.9%)	263 (2.9%)		263.3 (2.9%)	263.0 (2.9%)	
High-income	573,754 (5.5%)	548 (6.0%)		548.2 (6.0%)	548.0 (6.0%)	
Unknown	147,241 (1.4%)	125 (1.4%)		125.0 (1.4%)	125.0 (1.4%)	
**Marital status** **(not married *), *n* (%)**	6,958,660 (67.1%)	4866 (53.6%)	0.279	4869.2 (53.7%)	4866.0 (53.6%)	0.001
**Healthcare region, *n* (%)**			0.082			<0.001
North	899,110 (8.7%)	894 (9.9%)		894.3 (9.9%)	894.0 (9.9%)	
Stockholm	2,452,115 (23.7%)	1920 (21.2%)		1919.5 (21.2%)	1920.0 (21.2%)	
Southeast	1,076,468 (10.4%)	909 (10.0%)		908.7 (10.0%)	909.0 (10.0%)	
South	1,886,581 (18.2%)	1582 (17.4%)		1580.9 (17.4%)	1582.0 (17.4%)	
Uppsala-Örebrö	2,123,167 (20.5%)	1917 (21.1%)		1917.1 (21.1%)	1917.0 (21.1%)	
West	1,925,519 (18.6%)	1851 (20.4%)		1849.9 (20.4%)	1851.0 (20.4%)	

* Not married includes widow[er]s.

**Table 5 viruses-16-00271-t005:** Frequency of outcomes and case fatality rate for solid organ transplant recipients (SOTRs), unweighted comparison cohort (i.e., Swedish general population), and weighted comparison cohorts. Percentages are calculated from the total population at risk.

		SOTRs Overall vs. Unweighted Comparison Cohort	SOTRs Overall vs. Weighted Comparison Cohort	KTx vs. Weighted Comparison Cohort	LTx vs. Weighted Comparison Cohort	HTx vs. Weighted Comparison Cohort	LuTx vs. Weighted Comparison Cohort
**Total count in SOTR and weighted comparison cohorts**	SOTRs	9073	9073	5967	1866	778	462
	Comparators	10,362,960	9070.49	5967.54	1865.81	777.64	461.99
**Test-positive *n* (%)**	SOTRs	1204 (13.27%)	1204 (13.27%)	834 (13.98%)	215 (11.52%)	100 (12.85%)	55 (11.90%)
	Comparators	1,276,293 (12.32%)	1098.43 (12.11%)	732.83 (12.28%)	212.64 (11.40%)	91.36 (11.75%)	55.27 (11.96%)
**Infection *n* (%)**	SOTRs	1328 (14.64%)	1328 (14.64%)	915 (15.33%)	240 (12.86%)	108 (13.88%)	65 (14.07%)
	Comparators	1,294,658 (12.49%)	1150.90 (12.69%)	765.33 (12.82%)	226.02 (12.11%)	95.56 (12.29%)	57.66 (12.48%)
**Hospitalization *n* (%)**	SOTRs	628 (6.92%)	628 (6.92%)	447 (7.49%)	96 (5.14%)	46 (5.91%)	39 (8.44%)
	Comparators	74,245 (0.72%)	218.45 (2.41%)	143.52 (2.41%)	50.56 (2.71%)	15.95 (2.05%)	9.46 (2.05%)
**ICU care *n* (%)**	SOTRs	106 (1.17%)	106 (1.17%)	80 (1.34%)	11 (0.59%)	4 (0.51%)	11 (2.38%)
	Comparators	8101 (0.15%)	22.85 (0.25%)	16.59 (0.28%)	4.92 (0.26%)	1.66 (0.21%)	0.85 (0.18%)
**30-day mortality**	SOTRs	80 (0.88%)	80 (0.88%)	49 (0.82%)	15 (0.80%)	9 (1.16%)	7 (1.52%)
**(all-cause) *n* (%)**	Comparators	15,091 (0.17%)	65.37 (0.72%)	38.27 (0.64%)	15.56 (0.83%)	5.03 (0.65%)	3.12 (0.68%)
**60-day mortality**	SOTRs	98 (1.08%)	98 (1.08%)	59 (0.99%)	18 (0.96%)	12 (1.54%)	9 (1.95%)
**(all-cause) *n* (%)**	Comparators	18,004 (0.17%)	78.19 (0.86%)	45.67 (0.77%)	18.92 (1.01%)	6.07 (0.78%)	3.71 (0.80%)
**COVID-19-related death *n* (%)**	SOTRs	102 (1.12%)	102 (1.12%)	61 (1.02%)	20 (1.07%)	11 (1.41%)	10 (2.16%)
	Comparators	16,582 (0.16%)	71.01 (0.78%)	41.94 (0.70%)	16.62 (0.89%)	5.44 (0.70%)	3.40 (0.74%)
**Case fatality rate * (%)**	SOTRs	7.7%	7.7%	6.7%	8.3%	10.2%	15.4%
	Comparators	1.3%	6.2%	5.5%	7.4%	5.7%	5.9%

* Case fatality rate (CFR) was calculated using COVID-19-related death as the numerator and infection as the denominator. KTx: Kidney Transplanted, LTx: Liver Transplanted, HTx: Heart Transplanted, LuTx: Lung Transplanted.

**Table 6 viruses-16-00271-t006:** Hazard ratios (HR) for outcomes for solid organ transplant recipients (SOTRs) compared with weighted comparison cohorts.

Group	Outcome	HR	95% Confidence Interval	*p*
**SOTRs Overall**	Test-positive	1.09	1.03–1.15	0.004
	Infection	1.15	1.09–1.22	<0.001
	Hospitalization	2.89	2.67–3.13	<0.001
	ICU admission	4.59	3.76–5.60	<0.001
	30-day mortality (all-cause)	1.21	0.97–1.51	0.094
	60-day mortality (all-cause)	1.24	1.01–1.51	0.037
	COVID-19-related death	1.42	1.17–1.72	<0.001
**KTx**	Test positive	1.14	1.06–1.22	<0.001
	Infection	1.21	1.13–1.29	<0.001
	Hospitalization	3.16	2.88–3.47	<0.001
	ICU admission	4.80	3.83–6.02	<0.001
	30-day mortality (all-cause)	1.27	0.96–1.68	0.096
	60-day mortality (all-cause)	1.28	0.99–1.66	0.058
	COVID-19-related death	1.44	1.12–1.86	0.004
**LTx**	Test positive	0.98	0.85–1.12	0.709
	Infection	1.03	0.91–1.17	0.648
	Hospitalization	1.86	1.52–2.27	<0.001
	ICU admission	2.17	1.19–3.97	0.012
	30-day mortality (all-cause)	0.94	0.56–1.56	0.796
	60-day mortality (all-cause)	0.92	0.58–1.47	0.732
	COVID-19-related death	1.17	0.75–1.81	0.491
**HTx**	Test positive	1.08	0.89–1.32	0.431
	Infection	1.13	0.93–1.36	0.223
	Hospitalization	2.89	2.16–3.85	<0.001
	ICU admission	2.38	0.89–6.35	0.084
	30-day mortality (all-cause)	1.77	0.92–3.40	0.088
	60-day mortality (all-cause)	1.95	1.11–3.43	0.021
	COVID-19-related death	1.99	1.10–3.60	0.022
**LuTx**	Test positive	1.07	0.82–1.40	0.599
	Infection	1.23	0.96–1.57	0.101
	Hospitalization	4.52	3.30–6.19	<0.001
	ICU admission	13.82	7.64–25.0	<0.001
	30-day mortality (all-cause)	2.40	1.14–5.04	0.021
	60-day mortality (all-cause)	2.59	1.34–4.98	0.004
	COVID-19-related death	3.15	1.69–5.87	<0.001

KTx: Kidney Transplanted, LTx: Liver Transplanted, HTx: Heart Transplanted, LuTx: Lung Transplanted.

**Table 7 viruses-16-00271-t007:** Days from infection (test or diagnosis) to event (hospitalization, ICU-admission, 30-day mortality (all-cause), and COVID-19-related death) in solid organ transplant recipients (SOTRs), organ recipient subgroups and corresponding weighted comparison cohorts, presented as median (interquartile range).

Days from COVID-19 Infection to Respective Outcome	SOTRs Overall	Weighted Comparison Cohort for SOTRs	KTx	Weighted Comparison Cohort for KTx	LTx	Weighted Comparison Cohort for LTx	HTx	Weighted Comparison Cohort for HTx	LuTx	Weighted Comparison Cohort for LuTx
**Hospitalization**	0 (0–5)	0 (0–3)	0 (0–6)	0 (0–4)	0 (0–1.5)	0 (0–2)	0 (0–1)	0 (0–3)	0 (0–3)	0 (0–3)
**ICU admission**	7 (3–12)	4 (1–8)	8 (2–12)	4 (1–8)	6 (2–9)	4 (1–8)	3 (2–13)	5 (1–8)	7 (4–10)	4 (1–8)
**30-day mortality (all-cause)**	12 (6–21)	10 (6–16)	12 (6–22)	10 (6–16)	12 (6–22)	10 (6–17)	6 (3–10)	10 (6–17)	16 (11–21)	10 (6–16)
**60-day mortality (all-cause)**	14 (7–24)	11 (6–20)	14 (7–28)	11 (6–20)	13.5 (7–22)	11 (6–21)	9 (3.5–30)	11 (6–21)	19 (14–23)	11 (6–20)
**COVID-19-related death**	16 (7–32)	10 (5–18)	19 (9–33)	10 (6–18)	15 (7–50)	10 (5–18)	8 (3–29)	10 (6–19)	20 (14–64)	10 (5–18)

KTx: Kidney Transplanted, LTx: Liver Transplanted, HTx: Heart Transplanted, LuTx: Lung Transplanted.

## Data Availability

The individual-level data used in this study are pseudonymized and sourced from Swedish healthcare registers. Interested researchers can obtain access to the data from the appropriate Swedish public data holders, subject to obtaining ethics approval for the research and adhering to all relevant legislation, processes, and data protection protocols.

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
