# Peer review of "COVID-19 Outcomes and Vaccinations in Swedish Solid Organ Transplant Recipients 2020–2021: A Nationwide Multi-Register Comparative Cohort Study"

_viruses, 2024, doi:10.3390/v16020271_

Round 1

Reviewer 1 Report

Comments and Suggestions for Authors

This study is comprehensively done and well described. The authors show that morbidity and mortality was increased in solid organ transplant recipients (SOTRs), with lung transplant recipients showing the worst outcomes. They report nationwide data from Sweden for infections, hospitalisation, ICU-care and mortality from COVID-19 and vaccination data from 2020-2021. The analyses has been conducted in a rigorous and thorough manner, with careful selection of data (excluding potential database errors), use of comparison cohorts which were weighted based on a propensity score built from potential confounders to balance the SOTR and non-SOTR groups, and appropriate statistical tests for comparisons of the impact of COVID-19 on SOTR versus non-SOTR groups. 

Moreover, this study has significant benefit to the infectious diseases field - particularly for the consideration of appropriate public health measures and vaccine trials to protect vulnerable groups such as SOTRs. This study shows that SOTRs were poorly protected from vaccination and that COViD-19 vaccine trials contained insufficient data for SOTRs, which should be changed for future trials. Moreover, government guidelines had discouraged serological testing- this should be reconsidered in future pandemics as it may provide a means for early intervention in vulnerable groups with poor vaccination responses. This study also highlights that SOTRs showed an increased rate of infection compared with non-SOTRs despite being recommended to self-isolate. This highlights the need for more close monitoring of vulnerable groups for future pandemics and vaccination and routine testing for health workers that are in contact with vulnerable groups. 

Minor comments to be addressed:

- For the weighted comparisons (Table 4), what is the relevance of considering: i) marital status  and ii) education level ?

- The authors don't clear discuss the impact of the different waves of variants of SARS-CoV-2 on SOTRS vs non-SOTRS outcomes (Figure 1). Was there really no impact of the different variants on patient outcomes? Was it really just the poor vaccination response of SOTRs versus non-SOTRs that results in the divergence of the hospital and ICU admissions curves between these two groups? And how can you prove that is indeed the case?

Author Response

Thanks for your kind comments. Please find our response to you feedback below.

For the weighted comparisons (Table 4), what is the relevance of considering: i) marital status  and ii) education level ?

Marital status and education level were used in the matching as surrogates for other factors that may be confounding. We chose to use marital status as a surrogate for living in a household with members n>1 since it is less effective to enforce shielding practices when family members interact with other people through school or work. (Schools were not closed in Sweden, remote working was relatively uncommon and most work places were still in operation throughout the pandemic). Education level along with income were chosen as surrogates for social status/class since it has been shown in other works that lower social status was correlated with worse Covid-19 outcomes. We did not feel that income alone was adequate since this could be influenced by changes in pay after a period of disease and the transplant procedure.   

The authors don't clear discuss the impact of the different waves of variants of SARS-CoV-2 on SOTRS vs non-SOTRS outcomes (Figure 1). Was there really no impact of the different variants on patient outcomes? Was it really just the poor vaccination response of SOTRs versus non-SOTRs that results in the divergence of the hospital and ICU admissions curves between these two groups? And how can you prove that is indeed the case?

This is a great comment and one we grappled with a great deal when preparing the manuscript. The question is whether variant type leads to different outcomes in SOTRs compared with controls. There is very little evidence to support this hypothesis - especially on the variants circulating 2020-2021. There is however a great deal of evidence supporting worse vaccine response in SOTRs. We removed a section discussing the influence of variants since we felt it was too speculative and not adequately supported by other work.  

Reviewer 2 Report

Comments and Suggestions for Authors

I read with interest the paper entitled “COVID-19 Outcomes and Vaccinations in Swedish Solid Organ Transplant Recipients 2020–2021: A Nationwide Multi-Register Comparative Cohort Study”.

The authors conducted a nationwide cohort study encompassing all Swedish residents to examine the impact of SOTR on SARS-CoV2 infections hospitalization, ICU-care and mortality from COVID-19.

SOTRs had increased risk of contracting COVID-19, being hospitalized, receiving ICU-care, and dying.

This manuscript is well written, and the authors address an interesting research question. The data are clearly presented, and I did not find any significant methodological issues. Introduction and discussion are logically written. The manuscript does not require extensive English editing.

Minor points:

Table 3 and 4 – provide median of Charlson comorbidity index instead of frequencies.

Figure 1 and 2 - describe in more detail data in the text of manuscript. Figure 2 is hard to read.

Add conclusion paragraph (2-3 sentences) with main findings since the paper is quite long with lot of data.  

Author Response

Thank you foryour kind comments. Please find responses to your feedback below.

Table 3 and 4 – provide median of Charlson comorbidity index instead of frequencies.

Medians have been added at the top of each column. We chose this variant rather than replacement since the information provided in medians has very low "resolution".

Figure 1 and 2 - describe in more detail data in the text of manuscript. Figure 2 is hard to read.

We apologize for Figure 2 being hard to read when zoomed in. A new Figure 2 has been uploaded with higher resolution. The findigs presented in Figure 1 and 2 are described in section 3.2.2, 3.2.3 and 3.2.4. The findings are also supported by Table 5-7.

Add conclusion paragraph (2-3 sentences) with main findings since the paper is quite long with lot of data. 

A conclusion paragraph has been added which summarizes the most important findings

Reviewer 3 Report

Comments and Suggestions for Authors

I congratulate the authors of the manuscript that I had the pleasure of reading.
From the title to the conclusions, it is clear, engaging, and well-written.
The language is excellent, as is the presentation of the content.

The background is intriguing. Despite being concise, it provides essential elements to continue reading. I appreciate the authors' choice not to be overly prosaic. 

The methodology is well-described and robust. 

The results are clear and comprehensively presented. I would only suggest including Fig.2 in the supplementary materials: it is not easily readable (also due to the necessary enlargement it requires) and weighs down the reading. This is just my opinion, which I share with the authors.

The discussion is extremely interesting and impeccable. 

Congratulations once again!

Author Response

Thank you for your kind and encouraging comments!

The results are clear and comprehensively presented. I would only suggest including Fig.2 in the supplementary materials: it is not easily readable (also due to the necessary enlargement it requires) and weighs down the reading. This is just my opinion, which I share with the authors.

We have a submitted a higher resolution Fig.2 to make it more easy to read. We agree that it is not ideal to have so many panels in a Figure and experimented with different layouts during the preparation of this manuscript. The current iteration is the "least bad". We chose not to move it to supplementary materials since we feel it is the most informative part of the article, and it's shortcomings a somewhat offset by publishing in an online-only journal where readers can easily zoom in.  

Reviewer 4 Report

Comments and Suggestions for Authors

Comments on the manuscript titled: COVID-19 Outcomes and Vaccinations in Swedish Solid Organ 2 Transplant Recipients 2020–2021: A Nationwide Multi-Register 3 Comparative Cohort Study"

1. Abstract: The manuscript lacks a conclusion, and the objectives are not clearly defined.
2. The abstract should include the results of excess mortality. Effect sizes with their corresponding 95% confidence intervals (95%CI) should be presented in both the abstract and results section, with less emphasis on p-values.

3. Methods: To ensure comprehensive coverage of information in the secondary analysis manuscript, please reference the STROBE and RECORD checklists. Include the checklist as a supplementary table. Additionally, incorporate any missing information into the manuscript.

For reference, the RECORD checklist can be found here: https://www.record-statement.org/Files/checklist/RECORD%20Checklist.pdf

4. Figure Legend: Ensure that abbreviations used in the figures are explained in the corresponding legends.

5. The y-axis should not use abbreviation

6. The manuscript lacks a conclusive section, which should be addressed.

Author Response

Abstract: The manuscript lacks a conclusion, and the objectives are not clearly defined. The abstract should include the results of excess mortality. Effect sizes with their corresponding 95% confidence intervals (95%CI) should be presented in both the abstract and results section, with less emphasis on p-values.

Since this is a whole-population study (n=10,372,033) , absolute numbers have been presented in Table 5 and 95%CI is not relevant. Outcome HRs for SOTRs vs weighted comparators are presented in Table 6. 

Methods: To ensure comprehensive coverage of information in the secondary analysis manuscript, please reference the STROBE and RECORD checklists. Include the checklist as a supplementary table. Additionally, incorporate any missing information into the manuscript.

For reference, the RECORD checklist can be found here: https://www.record-statement.org/Files/checklist/RECORD%20Checklist.pdf

This was an excellent suggestion. The checklist has been filled in and uploaded

Figure Legend: Ensure that abbreviations used in the figures are explained in the corresponding legends.

Thank you for pointing out this oversight. It has been rectified.

The y-axis should not use abbreviation

The y-axis in Figure 2 uses the abbreviations COVID-19 and ICU. In our experience these two acronyms are among the few that are not written in full in medical literature. Therefore they are not written out in the main text body, tables or figures. 

The manuscript lacks a conclusive section, which should be addressed.

Thank you for the suggestion. A short conclusion paragraph has been added (section 5)

Round 2

Reviewer 4 Report

Comments and Suggestions for Authors

The authors did not fully addressed my comments.

1> The abstract is the most important part of the manuscript. There is no conclusion in the Abstract

1b> the objectives in the abstract is not clear, what is it?

1c> The abstract should include the results of excess mortality. Effect sizes with their corresponding 95% confidence intervals (95%CI) should be presented in both the abstract and results section, with less emphasis on p-values.The confidence interval is more important than the p-value. Please check the stat guideline. Readers don't need to see the tables and figures to understand the study.

2> I cannot see any revision in the methods that cite and indicate the study follow the guidlines and provide STROBE and RECORD checklists.

3> What is SMD in the figure of Appendix Table A3 (A-E)?

Author Response

1> The abstract is the most important part of the manuscript. There is no conclusion in the Abstract

The conclusion of the abstract has been changed to reflect the conclusion of the manuscript.

1b> the objectives in the abstract is not clear, what is it?

The objectives are stated in the third sentence of the abstract.

1c> The abstract should include the results of excess mortality. Effect sizes with their corresponding 95% confidence intervals (95%CI) should be presented in both the abstract and results section, with less emphasis on p-values.The confidence interval is more important than the p-value. Please check the stat guideline. Readers don't need to see the tables and figures to understand the study.

The absolute numbers for mortality have been added to the manuscript body due to the reviewers request. Presenting effect sizes in a weighted comparator group based on over ten million individuals is misleading to the reader. Presenting 95%CI for an absolute result (ie - documented mortality) in an entire population is not statistically possible but have been presented for Hazard ratios in Table 6. Adding any of the requested metrics to the abstract is confusing at best and misleading at worst. 

2> I cannot see any revision in the methods that cite and indicate the study follow the guidlines and provide STROBE and RECORD checklists.

This has been added to the methods section and checklists uploaded as supplementary materials.

3> What is SMD in the figure of Appendix Table A3 (A-E)?

The standardized mean difference (SMD) of covariates is defined in the Table and Figure legend of Appendix Table A3 (A-E)